# Improved Optical Efficiency of 850-nm Infrared Light-Emitting Diode with Reflective Transparent Structure

**DOI:** 10.3390/mi14081586

**Published:** 2023-08-12

**Authors:** Hyung-Joo Lee, Jin-Young Park, Lee-Ku Kwac, Jongsu Lee

**Affiliations:** 1CF Technical Division, AUK Corporation, Iksan 54630, Republic of Korea; dieblood77@nate.com; 2Korea Photonic Technology Institute (KOPTI), Gwangju 61007, Republic of Korea; police7630@kopti.re.kr; 3Department of Carbon Convergence Engineering, Jeonju University, Jeonju 55069, Republic of Korea; 4Department of Advanced Components and Materials Engineering, Sunchon National University, Suncheon 57922, Republic of Korea

**Keywords:** transparent structure, reflector structure, infrared, light-emitting diode, light photon

## Abstract

This study investigated a reflective transparent structure to improve the optical efficiency of 850 nm infrared light-emitting diodes (IR-LEDs), by effectively enhancing the number of extracted photons emitted from the active region. The reflective transparent structure was fabricated by combining transparent epitaxial and reflective bonding structures. The transparent epitaxial structure was grown by the liquid-phase epitaxy method, which efficiently extracted photons emitted from the active area in IR-LEDs, both in the vertical and horizontal directions. Furthermore, a reflective bonding structure was fabricated using an omnidirectional reflector and a eutectic metal, which efficiently reflected the photons emitted downwards from the active area in an upward direction. To evaluate reflective transparent IR-LED efficiency, a conventional absorbing substrate infrared light-emitting diode (AS IR-LED) and a transparent substrate infrared light-emitting diode (TS IR-LED) were fabricated, and their characteristics were analyzed. Based on the power–current (L-I) evaluation results, the output power (212 mW) of the 850 nm IR-LED with the reflective transparent structure increased by 76% and 26%, relative to those of the AS IR-LED (121 mW) and TS IR-LED (169 mW), respectively. Furthermore, the reflective transparent structure possesses both transparent and reflective properties, as confirmed by photometric and radial theta measurements. Therefore, light photons emitted from the active area of the 850 nm IR-LED were efficiently extracted upward and sideways, because of the reflective transparent structure.

## 1. Introduction

Near-infrared light-emitting diodes (NIR-LEDs) are commonly utilized as emitters for photo-couplers, automobile sensors, and closed-circuit television [1,2]. In recent years, they have been applied on a wide scale, including in time-of-flight sensors, optical sensors used in wearable devices, small vehicles, and flying drones [3,4]. Smaller NIR-LEDs with higher output power at large injection currents are required for certain applications. The output power for NIR-LEDs can be increased by using multiple quantum wells (MQWs), window layers, distributed Bragg reflectors (DBRs), omnidirectional reflectors (ODRs), and current-spreading layers. MQWs are used to maximize the internal quantum efficiency of the active region in NIR-LEDs [5]. To improve the optical efficiency of NIR-LEDs with an absorbing substrate, a DBR must be used because it reflects the photons emitted from the active area in an upward direction [6]. To obtain improved reflectivity and thermal dissipation efficiency, a reflective single metal and eutectic metal have been used. These metals serve to upwardly reflect photons emitted downward to the active area or to dissipate significant heat formed in the active area [7]. Additionally, photons absorbed by the top electrode can easily escape from the LED through thick sideway paths produced by using a current-spreading layer [8]. These studies emphasize that the optical efficiency must be improved by reflecting or moving photons emitted from the active areas in a light-emitting diode (LED). However, an alternative solution for a sharp decrease in chip size has not yet been proposed. Several studies indicate high size-dependent efficiency, in which smaller devices exhibit lower maximum efficiency, attributed to the degradation of electrical injection [9,10]. A sidewall passivation treatment and Si substrate were employed to overcome the size-drop effect for several microdevices. High light output power, high size-independent leakage current density, and low ideality factor were observed by employing sidewall treatments [11]. In smaller devices, the silicon substrate was more effective than the GaAs substrate, owing to the former’s thermal dissipation effect [12]. However, conventional studies do not address the sudden decrease in the surface and sideway emission areas of these devices caused by a reduction in the chip size.

In this study, we focused on improving the optical path capability of the surface and side emissions, which sharply deteriorated because of the reduction in the chip size. Here, transparent and reflective structures are selected and investigated as candidates for solving the abovementioned problems because of their proven success in enhancing the output power of LEDs [13,14]. Owing to the transparent structure of the IR-LED, a significant number of light photons emitted from the active region may be induced to emit sideways. Conversely, the use of a reflective structure fabricated by the wafer-bonding process may effectively increase the number of light photons emitted upward from the active region.

Furthermore, exploiting the advantages of both approaches is an effective solution to improve the optical path capability of both surface and sideway emissions. Therefore, we verified the applicability of a combination of the transparent epitaxial and reflective bonding structures toward addressing the aforementioned problems. 

The transparent epitaxial structure in the IR-LED was obtained using thick *p*- and *n*-AlGaAs layers, grown using the liquid-phase epitaxy (LPE) method. A reflective bonding structure was applied using a reflector/eutectic bonding process for an IR-LED with a transparent structure. As a result, a remarkably improved output power was observed for the IR-LED chip with this developed combined structure, in relation to those of other IR-LED chips. Therefore, this study verified that a combined structure is crucial for improving the output power of IR-LEDs. 

## 2. Materials and Methods

Using the metalorganic chemical vapor deposition (MOCVD) or LPE method, epitaxial wafers with wavelengths of 850 nm were created to produce the developed samples. By using the conventional 850 nm epitaxial wafer (LED A) fabricated via MOCVD, five pairs of MQWs, each with 5 nm thick GaAs wells and 12 nm thick Al_0.05_Ga_0.95_As barriers, were used as the active region. The *n*- and *p*-type confinement layers made of *n*- and *p*-doped Al_0.3_Ga_0.7_As materials, respectively, were on each side of the active area. The *n*- and *p*-doped Al_0.3_Ga_0.7_As materials were 2.5 × 10^18^ atoms/cm^−3^ and 1.5 × 10^18^ atoms/cm^−3^, respectively. The *n*-doped Al_0.12_Ga_0.88_As/*n*-Al_0.9_Ga_0.1_As material (high refraction index: 60 nm/low refraction index: 70 nm) was inserted between *n*-doped Al_0.3_Ga_0.7_As and *n*-type GaAs substrates in the 20 paired DBRs of the LED A structure. For the LED B and LED C structures fabricated using the LPE method, 120 μm thick 2nd *n*-Al_0.6_Ga_0.4_As, and 20 μm thick 1st *n*-Al_0.18_Ga_0.82_As layers grown sequentially on the GaAs substrate were employed as the etching stop layer and *n*-confinement, respectively. Moreover, 1-μm-thick p-Al_0.08_Ga_0.92_As and 20-μm-thick 1st *n*-Al_0.18_Ga_0.82_As were grown on *n*-confinement as the active layer and p-confinement, respectively. Additionally, LEDs with transparent and reflective structures must be fabricated through LPE at an epitaxial growth rate of 1 µm/min. 

Before the wafer-bonding process, the absorbing GaAs substrate was selectively removed from the H_2_O_2_:NH_3_ solution until the appearance of the 2nd *n*-Al_0.6_Ga_0.4_As layer, which was attached to a p-Si carrier by a paraffin solid. After removing the n-GaAs substrate, the p-Si carrier was selectively removed by eliminating the paraffin. Thus, the transparent substrate LED (TS LED) structure was obtained (LED B). 

An epitaxial wafer (LED B) without a GaAs substrate was wafer-bonded to the p-Si substrate. A 3000 nm thick Ti/Au/In/Ti structure was employed as the eutectic structure, while a reflector made of 500 nm thick Ag was used for the reflector to bond the wafers. A pressing force of 4500 N at 230 °C was used to conduct the wafer-bonding process for 60 min. Therefore, a reflective transparent substrate LED (reflective TS IR-LED) was finally obtained (LED C). Figure 1 shows the fabrication process for three types of infrared (IR) LEDs: absorbing substrate 850 nm IR-LED (LED A), transparent substrate 850 nm IR-LED (LED B), and reflective transparent 850 nm IR-LED (LED C). The absorbing substrate 850 nm IR-LED (LED A) was grown in situ on an n-GaAs absorbing substrate, using the MOCVD system. The 850 nm IR-LEDs with the transparent substrate were grown on an n-GaAs absorbing substrate using the liquid-phase epitaxy system. The TS 850 nm IR-LED was obtained simply by removing the n-GaAs absorbing substrate. The reflective TS 850 nm IR-LED was fabricated by adding a reflective structure (reflector/eutectic/p-Si) to the TS 850 nm IR-LED. It is important to note that the reflective TS 850 nm IR-LED should have a reverse structure.

Bonded IR-LED wafers were sequentially cleaned with acetone and methanol to remove organic contamination, followed by removing the surface oxidation of the 2nd n-Al_0.6_Ga_0.4_As top window (front) and p-Si substrate (back) in an HF:deionized water (10:1) solution. After cleaning, bonding pads were placed on the front and back of the wafers using a combination of photolithography and selective etching. AuGeNi (1000 nm/50 nm/20 nm) was deposited on the n-type substrate using a thermal evaporator, and AuBe (500 nm) was deposited on the p-type substrate using an electron beam evaporator. Figure 2 shows the schematic of the structure, and provides the compositional information of the reflective TS 850 nm IR-LED chip. 

## 3. Results and Discussion

Previous studies showed that the emission area must be increased to improve the efficiency of the IR-LEDs. In this study, we demonstrate that the combined structure of the transparent and reflective layers can serve as an effective light-emitting factor for IR-LEDs. Conventional IR-LEDs have been fabricated through MOCVD with a low growth rate (1 µm/h). Therefore, we conducted LPE to obtain IR-LEDs with transparent structures. Through LPE, tens-of-micrometer-thick transparent structures could be grown in the IR-LED with a noticeable growth rate (1 µm/min). Figure 3a,b show the epitaxial scanning electron microscopy (SEM) images of the conventional IR-LED and TS LED. The conventional IR-LED in Figure 3a exhibits the total epitaxial layers of approximately 6 µm thick, which include the DBR used as the reflector. Based on existing research, the number of light photons emitted from the active region decreases upon the reduction of the emission area caused by a thin epitaxial layer. However, the conventional IR-LED could not have a thick transparent layer, owing to its inefficient growth rate. In the SEM image in Figure 3b, a significantly thick transparent layer (p-, n- AlGaAs) is observed in the active region (p-Al_0.08_Ga_0.92_As). The n-Al_0.6_Ga_0.4_As layer used for the 2nd n-confinement was approximately 120 µm thick. The p-AlGaAs and n-AlGaAs layers at the interface of the active region were approximately 20 µm thick. Furthermore, a reflective bonding structure was used to increase the emission area of the developed TS IR-LED. Figure 3c shows the SEM image of the epitaxial layer of the reflective TS IR-LED, fabricated using the wafer-bonding process. The SEM image in Figure 3c shows the reversed TS structure, reflective bonding structure, and the p-Si layer. Figure 3d–f show the schematic of the structure of the chip fabricated from the epitaxial structures in Figure 3a–c. In the conventional IR-LED chip shown in Figure 3d, the active region between the p- and n-confinements is located on the DBR and absorbing GaAs substrate. Contrarily, the TS IR-LED chip has no DBR or absorbing GaAs substrate, except for the transparent p- and n-AlGaAs layers. The reflective TS IR-LED chip in Figure 3f has a reversed TS epitaxial layer on the reflective bonding structure and a p-Si substrate applied by the wafer-bonding process. Based on the optical analysis, the optical-emission efficiency of the reflective TS IR-LED chip will be higher than that of either the AS IR-LED chip or the TS IR-LED chip, as shown in Figure 3c,f.

Figure 4 shows the photon paths for the AS IR-LED chip with the DBR, the TS IR-LED chip with the transparent layer, and reflective TS IR-LED chips with both transparent and reflective layers. From the photon paths shown on the AS IR-LED chips in Figure 4a, most of the light emitted from the active region escaped from the LED chip through the surface. In addition, some light photons emitted downward from the active region may have escaped to the surface via the DBR. Despite these efforts, the AS IR-LED had a relatively low optical efficiency, owing to either an insignificant sideway emitting angle or an insignificant DBR reflective angle. The TS IR-LED structure in Figure 4b shows that a significant number of light photons could effectively escape from the LED chip by increasing the sideway emission area of the transparent layer. This remarkable improvement was reasonable because the emission area was much larger than that of the surface in the IR-LED. However, even with the use of the TS IR-LED chip, most light photons emitted downward from the active region did not escape from the LED chip, because either Ag paste or bonding metal was used for the assembly. Therefore, the reflective TS IR-LED chip in Figure 3c would exhibit a higher optical efficiency than the TS IR-LED chip. Here, Figure 4c shows that most of the light emitted downward from the active region can effectively escape from the LED chip when both a transparent layer and an ODR are used, owing to the significant reduction in the intensity of light moving downward. A significant reduction was achieved by enhancing the sideway-directed light and upwardly reflected light generated by the transparent and reflective layers, respectively.

The results of the light-emitting paths in Figure 4 verified that the use of transparent and reflective layers was a more attractive method for improving the light extraction efficiency of the IR-LED chip, because the light-emitting path in the LED chip was limited by the intrinsic problems of an insignificant sideway emission area and the low reflectivity of the specific wavelength [15]. The results in Figure 3 and Figure 4 demonstrate that the light extraction efficiency of IR-LEDs can be improved using either the transparent epitaxial layer or the reflective bonding layer.

To obtain more detailed information, the current–voltage *(I-V)* and light output power–current *(L-I)* characteristics of the AS IR-LED, TS IR-LED, and reflective TS IR-LED chips were evaluated (Figure 5). Here, an integrating sphere was used to measure the output power–current–voltage (*L*-*I*-*V*) characteristics of the developed LEDs. An integrating sphere is designed to collect light scattered and emitted from a sample in the form of a hollow sphere with a highly reflective inner surface (Model OPI-100 LED Electrical and, Optical Test System, Withlight company, Yeoju-si, Republic of Korea).

As shown in Figure 5a, a marginally higher turn-on voltage (0.1 V) of the AS IR-LED chip is induced by the resistance of the undoped active region in the AS IR-LED. The TS IR-LED and reflective TS IR-LED chips exhibited similar turn-on voltage properties because the series resistance of the device was not significantly influenced by the metals (used as a reflector or eutectic structure). However, the current–voltage curve exhibits a different trend with increasing current. When the current was increased, the reflective TS-IR LED chips showed a relatively lower rate of increase than that of the TS IR-LED chips. This may have been owing to the heat-dissipation effect caused by the metal (reflector/eutectic metal) used in the reflective TS IR-LED [16]. The output powers of the developed IR-LED chips exhibit different properties in Figure 5b. At an injection current of 300 mA, the TS IR-LED chip with a transparent layer exhibited a higher output power (169 mW) than the AS IR-LED chip (121 mW). Furthermore, an improved output power (202 mW) was obtained from the reflective TS IR-LED chip with both the transparent and reflective layers. This result confirmed that the output power of the IR-LED chips was strongly dependent on the use of either a transparent or reflective layer, because light photons emitted from the active region can effectively escape from the LED upward or sideways. Moreover, the results of the output power exhibited a trend similar to those of the light photon path illustrated in Figure 4. Therefore, the IR-LED chip with the combined structure would exhibit a considerably higher output power than those with either a transparent or reflective layer. 

In particular, the IR-LED chip with a combined reflector had a 67% higher output power than a conventional LED A with a DBR. Furthermore, the radial theta (half angle) of the photometric values was investigated for the AS IR-LED, TS IR-LED, and reflective TS IR-LED chips; the results are shown in Figure 6. Here, a light distributor (goniophotometer) was used to measure the radial theta of the photometric values for the developed LEDs. A light distributor is a piece of equipment that measures the intensity of light reflected from the surface of an object at various angles, as well as analyzing the direction and distribution characteristics of light from a light source, lighting fixture, medium, and surface. (Model OPI-305 Gonio-Photometer System, Withlight company, Republic of Korea). In the case of the conventional IR-LED chip (LED A) with DBRs, the radial theta was a relatively narrow angle, and the photometric value was low. A relatively high photometric value (68–70) was observed between 0° and ±10°. Above 20°, the photometric value exhibited a remarkably decreasing trend. Therefore, the DBRs were more effective in reflecting photons from the surface. However, the TS IR-LED chip with a transparent layer had a wider radial theta and higher photometric value. A higher photometric value (80–81) was observed between 0° and ±40°. Therefore, the photons escaped from the IR-LED sideways, because of the transparent layer. In the case of the reflective TS IR-LED chip, a higher photometric value (~100) was observed at similar angles (0°–38°). As a result, different light–current (L-I) curves of the AS IR-LED, TS IR-LED, and reflective TS IR-LED chips were obtained by using the radial theta and photometric values. 

These results demonstrated that reflective transparent structures are essential in decreasing the surface and sideway emission areas, owing to a sharp chip shrink. The reflective transparent structure exhibited a high efficiency in extracting photons emitted sideways from the active area. Conversely, the reflective structure efficiently reflected photons emitted from the active area in the upward direction. Therefore, the transparent structure is more useful for improving optical efficiency reduced by chip shrinking because of emission areas on the four sides. Furthermore, the reflective structure is one of the factors in improving optical efficiency. Therefore, much thicker transparent and reflector structures were crucial for improving optical efficiency of an extremely smaller chip. The mutual complementarity between the combined structure and wavelength must be considered for developing LEDs with shorter or longer wavelengths.

## 4. Conclusions

The optical efficiency of the 850 nm IR-LED was sharply decreased by a sudden decline in the surface and sideway emission areas of these devices caused by a drop in chip size. In this study, we demonstrated that optical efficiency could be sharply increased by increasing the surface and sideway emission areas through the use of the reflective transparent structure. This is because the surface and sideway emission areas were effectively respectively increased by the transparent and reflective structures in the LED bearing the reflective transparent structure. The transparent epitaxial substrate was secured by applying thick p- and n-AlGaAs epitaxial layers through the LPE method. A reflector bonding structure was sequentially fabricated on the transparent epitaxial structure by applying a reflector and eutectic structure. The presence of the reflective transparent structure in the 850 nm IR-LED was confirmed through SEM. From the schematic of photon paths, the highest optical extraction efficiency was observed from the IR-LED with a reflective transparent structure. This high efficiency was attributed to the enlargement of the effective upward and sideway emission areas caused by the reflective transparent structure. This theoretical result was supported by results of the output power–current–voltage (L-I-V) characteristic and relative photometric measurement. Based on the output power–current (L-I) curves, the IR-LED with reflective transparent structure was found to exhibit a 76% and 26% higher output power than the AS IR-LED and TS IR-LED, respectively. Furthermore, through photometric and radial theta measurements, the emission area of an IR-LED with a reflective transparent structure was much wider and straighter than those of the others. These results suggest that a reflective transparent structure combining the transparent epitaxial structure and reflector bonding structure may considerably increase the optical efficiency of 850 nm IR-LEDs.

## Figures and Tables

**Figure 1 micromachines-14-01586-f001:**
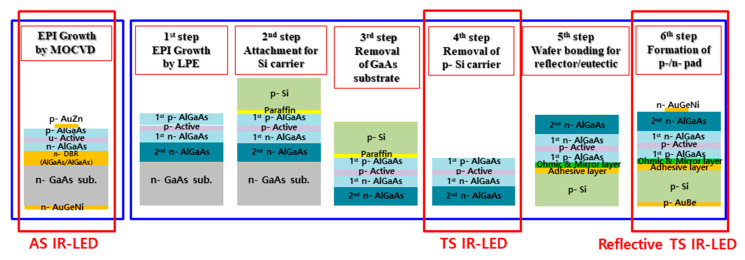
Fabrication process for developed AS, TS, and reflective TS IR-LEDs.

**Figure 2 micromachines-14-01586-f002:**
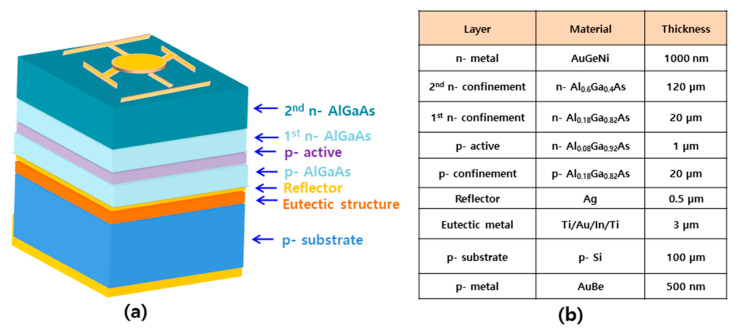
(**a**) Schematic of the structure. (**b**) Composition of reflective TS 850 nm IR-LED chip.

**Figure 3 micromachines-14-01586-f003:**
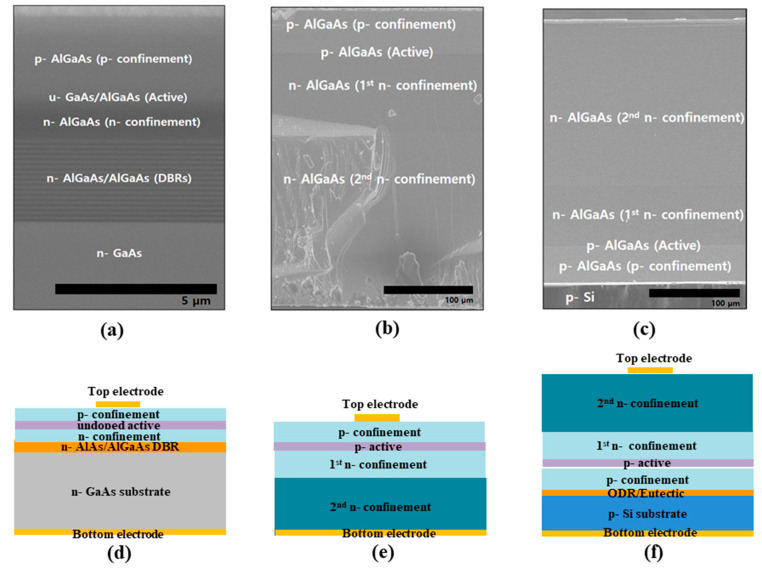
SEM images of the epitaxial layers (**a**–**c**) and schematic (**d**–**f**) of the structures of the AS IR-LED chip, TS IR-LED chip, and reflective TS IR-LED chip.

**Figure 4 micromachines-14-01586-f004:**
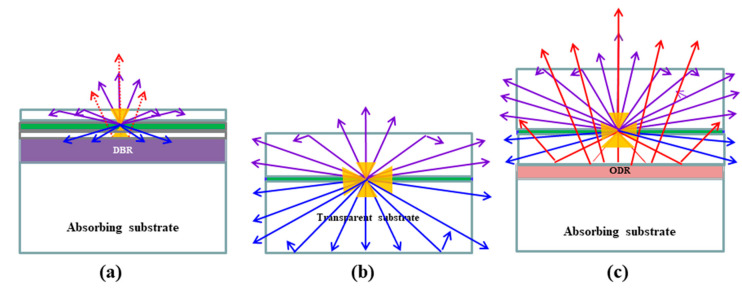
Photon paths of the (**a**) AS IR-LED chip, (**b**) TS IR-LED chip, and (**c**) reflective TS IR-LED chip.

**Figure 5 micromachines-14-01586-f005:**
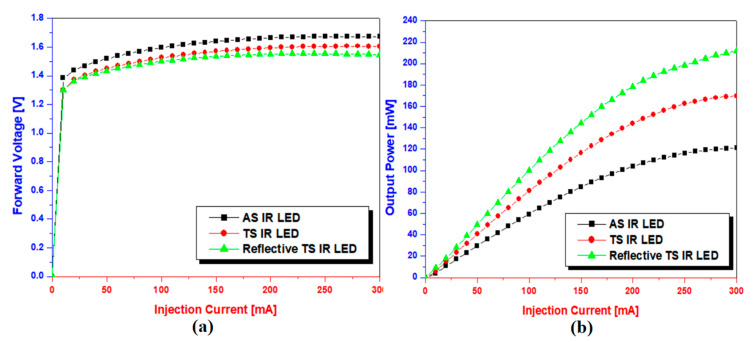
L-I-V curve for AS IR-LED chip, TS IR-LED chip, and Reflective TS IR-LED chip: (**a**) I-V curve and (**b**) L-I curve.

**Figure 6 micromachines-14-01586-f006:**
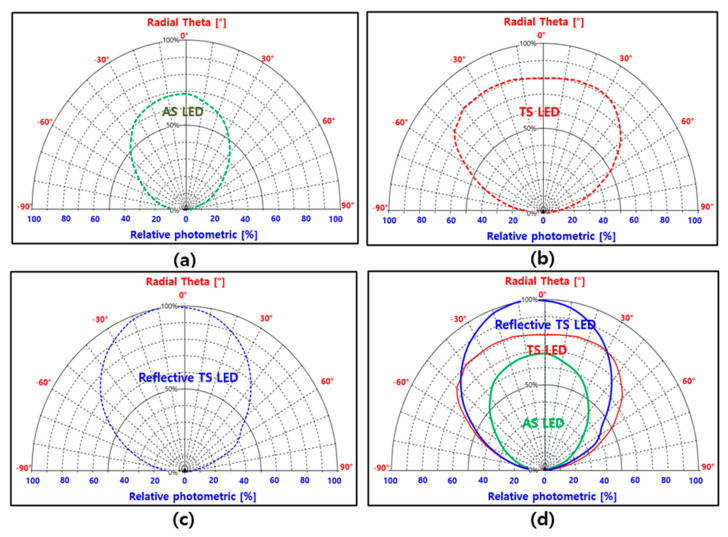
Relative photometric and radial theta (half angle) values of the (**a**) AS IR-LED chip, (**b**) TS IR-LED chip, (**c**) reflective TS IR-LED chip, and (**d**) integrated IR-LED chips.

## Data Availability

Not applicable.

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
