# Peer review of "Improved Optical Efficiency of 850-nm Infrared Light-Emitting Diode with Reflective Transparent Structure"

_micromachines, 2023, doi:10.3390/mi14081586_

Round 1
Reviewer 1 Report
In the current paper, an investigation of the optical characteristics of a novel diode light source is performed. A few minor modifications are suggested before proceeding to publication:
1) In the introduction, authors should minimize the number of bundled references. Instead, they should try to highlight particular aspects of other works in the literature that are directly related to the work proposed, whenever possible.
2) Related to the previous point, the introduction should clearly state what is the novelty of the current work compared to the previous knowledge. Authors should try to answer two questions: how is the proposed work different from previous studies? why is it important?
3) Figure 1 has a typographical error ("Romoval").
4) Was there more than one sample fabricated using the procedure described in Figure 1? If so, authors should discuss the repeatability of the process and how it affects the subsequent diode performance.
5) From the document it is not clear how the experiments related to Figures 5 and 6 were performed. Please indicate the methodology followed and characteristics of the sensors used
Author Response
Point-By-Point Response Letter (Micromachines-2527595)
In this response letter, we addressed all the comments made by the reviewers to our manuscript (Micromachines-2527595). All revisions made to the manuscripts addressing the comments of the reviewers are highlighted in the blue text.
Comments and Suggestions for Authors
In the current paper, an investigation of the optical characteristics of a novel diode light source is performed. A few minor modifications are suggested before proceeding to publication:
[Comment #1]
In the introduction, authors should minimize the number of bundled references. Instead, they should try to highlight particular aspects of other works in the literature that are directly related to the work proposed, whenever possible.
[Response]
The references cited in the introduction are crucial for explaining the relevance of the study. However, we separated the bundled references [5-8] and [9-12], and provided detailed information on each reference.
Revised manuscript (Line 40-58)
âž MQWs are used to maximize the internal quantum efficiency of the active region in NIR-LEDs [5]. To improve the optical efficiency of NIR-LEDs with an absorbing substrate, a DBR must be used because it reflects the photons emitted from the active area in an upward direction [6]. To obtain improved reflectivity and thermal dissipation efficiency, a reflective single metal and eutectic metal have been used. These metals serve to upwardly reflect photons emitted downward to the active area or to dissipate significant heat formed in the active area [7]. Additionally, photons absorbed by the top electrode can easily escape from the LED through thick sideway paths produced by using a current-spreading layer [8]. These studies emphasize that the optical efficiency must be improved by reflecting or moving photons emitted from the active areas in a light-emitting diode (LED). However, an alternative solution for a sharp decrease in chip size has not yet been proposed. Several studies indicate high size-dependent efficiency, in which smaller devices exhibit lower maximum efficiency attributed to the degradation of electrical injection [9,10]. A sidewall passivation treatment and Si substrate were employed to overcome the size-drop effect for several microdevices. High light output power, high size-independent leakage current density, and low ideality factor were observed by employing sidewall treatments [11]. In smaller devices, the silicon substrate was more effective than the GaAs substrate owing to the former’s thermal dissipation effect [12].
[Comment #2]
Related to the previous point, the introduction should clearly state what is the novelty of the current work compared to the previous knowledge. Authors should try to answer two questions: how is the proposed work different from previous studies? why is it important?
[Response]
Thank you for the suggestions. Please note that the revision has been made.
Revised manuscript (Lines 60~71)
âž In this study, we focused on improving the optical path capability of the surface and side emissions, which sharply deteriorated because of the reduction in the chip size. Here, transparent and reflective structures are selected and investigated as candidates for solving the abovementioned problems because of their proven success in enhancing the output power of LEDs [13,14]. Owing to the transparent structure of the IR-LED, a significant number of light photons emitted from the active region may be induced to emit sideways. Conversely, the use of a reflective structure fabricated by the wafer-bonding process may effectively increase the number of light photons emitted upward from the active region.
Furthermore, exploiting the advantages of both approaches is an effective solution to improve the optical path capability of both surface and sideway emissions. Therefore, in this study, we verified the applicability of a combination of the transparent epitaxial and reflective bonding structures toward addressing the aforementioned problems.
[Comment #3].
Figure 1 has a typographical error ("Romoval").
[Response]
We apologize for the oversight. The figure has been revised. (Romoval -> Removal)
Revised manuscript (Line 116)
âž Figure 1.
[Comment #4]
Was there more than one sample fabricated using the procedure described in Figure 1? If so, authors should discuss the repeatability of the process and how it affects the subsequent diode performance.
[Response]
Thank you for the suggestion. In the procedure depicted in Figure 1, there are transparent substrate LED (TS LED) and reflective transparent substrate LED (Reflective TS LED). It was marked in Fig 1.
Revised manuscript (Lines 106~ 116)
âž Figure 1 shows the fabrication process for three types of infrared (IR) LEDs: absorbing substrate 850 nm IR LED (LED A), transparent substrate 850 nm IR LED (LED B), and reflective transparent 850nm IR LED (LED C). The absorbing substrate 850 nm IR LED (LED A) was grown in-situ on an n-GaAs absorbing substrate using the MOCVD system. The 850 nm IR LEDs with the transparent substrate were grown on an n-GaAs absorbing substrate using the liquid phase epitaxy system. The TS 850 nm LED was obtained simply by removing the n-GaAs absorbing substrate. The reflective TS 850 nm LED was fabricated by adding a reflective structure (reflector/eutectic/p-Si) to the TS 850 nm LED. It's important to note that the reflective TS 850 nm LED should have a reverse structure.
[Comment #5]
From the document it is not clear how the experiments related to Figures 5 and 6 were performed. Please indicate the methodology followed and characteristics of the sensors used.
[Response]
Thank you for the comment. Our aim was to provide information regarding the L-I-V and relative photometric measurements. Please note that the necessary revision made in accordance with your comments.
Revised manuscript (Lines 201~205)
âž Here, an integrating sphere was used to measure the output power–current–voltage (L-I-V) characteristics of the developed LEDs. An integrating sphere is designed to collect light scattered and emitted from a sample in the form of a hollow sphere with a highly reflective inner surface [Model OPI-100 LED Electrical & Optical Test System, Withlight company, Republic of Korea].
Revised manuscript (Lines 230 ~ 235)
âž Here, a light distributor (Gonio-photometer) was used to measure the radial theta of the photometric values for the developed LEDs. A light distributor is an equipment that measures the intensity of light reflected from the surface of an object at various angles as well as analyzes the direction and distribution characteristics of light from a light source, lighting fixture, medium, and surface. [Model OPI-305 Gonio-Photometer System ,Withlight company, Republic of Korea].

Reviewer 2 Report
This manuscript has investigated the reflective transparent structure and improved the optical efficiency of 850-14 nm infrared light-emitting diodes. However, the manuscript has some defect. Thus, I suggest that the paper be major revision. The following hints may help the authors:
Q1: The writing and grammar should be extensively improved. The current version of the manuscript is hard to read because it is very poorly written. I encourage the authors to work with an English speaker in order to improve the readability of the text, especially tenses. Many grammatical errors were found.
Q2: In paper, please avoid the lump literature, such as [9-12],[5-8] and so on, summarize the main contribution of each references paper in separate sentences. The reference style should be checked again according to the journal standard.
Q3: There are a lot of abbreviations in the article, and the full names are not indicated. It is necessary for the author to add a nomenclature at the beginning or end of the article. Avoid using abbreviations in the Title, Abstract and Conclusions, if possible.
Q4: In the introduction, the reflective transparent structure should be introduced, which has demonstrated the necessity of the study. The author should not only provide a narrative of the results, but also have their own evaluation. Otherwise, the innovation of the paper and the necessity of research will not be perfectly displayed.
Q5: In general,the conclusion is not well organized. The results should be further elaborated to show how they could be used for the real applications.
Q6: Could you rewrite the abstract to be more focused on the main goal and the key obtained findings?
Please see the Comments and Suggestions for Authors.
Author Response
Point-By-Point Response Letter (Micromachines-2527595)
In this response letter, we addressed all the comments made by the reviewers to our manuscript (Micromachines-2527595). All revisions made to the manuscripts addressing the comments of the reviewers are highlighted in the blue text
Comments and Suggestions for Authors
This manuscript has investigated the reflective transparent structure and improved the optical efficiency of 850-14 nm infrared light-emitting diodes. However, the manuscript has some defect. Thus, I suggest that the paper be major revision. The following hints may help the authors
[Comment #1]
The writing and grammar should be extensively improved. The current version of the manuscript is hard to read because it is very poorly written. I encourage the authors to work with an English speaker in order to improve the readability of the text, especially tenses. Many grammatical errors were found.
[Response]
Thank you for the feedback. Please note that the manuscript was proofread by a professional editing company.
[Comment #2]
In paper, please avoid the lump literature, such as [9-12],[5-8] and so on, summarize the main contribution of each references paper in separate sentences. The reference style should be checked again according to the journal standard.
[Response]
The references in the introduction are crucial for explaining the relevance of the study. Therefore, we separated the combined references [5-8] and [9-12], and provided detailed information on each reference.
Revised manuscript (Line 40-58)
âž MQWs are used to maximize the internal quantum efficiency of the active region in NIR-LEDs [5]. To improve the optical efficiency of NIR-LEDs with an absorbing substrate, a DBR must be used because it reflects the photons emitted from the active area in an upward direction [6]. To obtain improved reflectivity and thermal dissipation efficiency, a reflective single metal and eutectic metal have been used. These metals serve to upwardly reflect photons emitted downward to the active area or to dissipate significant heat formed in the active area [7]. Additionally, photons absorbed by the top electrode can easily escape from the LED through thick sideway paths produced by using a current-spreading layer [8]. These studies emphasize that the optical efficiency must be improved by reflecting or moving photons emitted from the active areas in a light-emitting diode (LED). However, an alternative solution for a sharp decrease in chip size has not yet been proposed. Several studies indicate high size-dependent efficiency, in which smaller devices exhibit lower maximum efficiency attributed to the degradation of electrical injection [9,10]. A sidewall passivation treatment and Si substrate were employed to overcome the size-drop effect for several microdevices. High light output power, high size-independent leakage current density, and low ideality factor were observed by employing sidewall treatments [11]. In smaller devices, the silicon substrate was more effective than the GaAs substrate owing to the former’s thermal dissipation effect [12].
[Comment #3]
There are a lot of abbreviations in the article, and the full names are not indicated. It is necessary for the author to add a nomenclature at the beginning or end of the article. Avoid using abbreviations in the Title, Abstract and Conclusions, if possible.
[Response]
Thank you for pointing this out. The numbers of abbreviations were minimized in the abstract and conclusion. We have also inserted the nomenclature in the manuscript.
Revised manuscript (Lines 291 ~295)
âž Nomenclatures:
NIR-LED (near-infrared light-emitting diode), AS IR-LED (absorbing substrate infrared light-emitting diode), TS IR-LED (transparent substrate infrared light-emitting diode), MQWs (multiple quantum wells), DBRs (distributed Bragg reflector), ODR (omnidirectional reflector), LPE (liquid-phase epitaxy), MOCVD (metal-organic chemical vapor deposition), SEM (scanning electron microscopy).
[Comment #4]
In the introduction, the reflective transparent structure should be introduced, which has demonstrated the necessity of the study. The author should not only provide a narrative of the results, but also have their own evaluation. Otherwise, the innovation of the paper and the necessity of research will not be perfectly displayed.
[Response]
Please note that the revision has been made.
Revised manuscript (Lines 247 ~ 257)
âž These results demonstrated that reflective transparent structures are essential in decreasing the surface and sideway emission areas owing to a sharp chip shrink. The reflective transparent structure exhibited a high efficiency in extracting photons emit-ted sideways from the active area. Conversely, the reflective structure efficiently reflected photons emitted from the active area in the upward direction. Therefore, the transparent structure is more useful for improving optical efficiency reduced by chip shrinking because of emission areas on the four sides. Furthermore, the reflective structure is one of the factors to improve optical efficiency. Therefore, much thicker transparent and reflector structures were crucial for improving optical efficiency of an extremely smaller chip. The mutual complementarity between the combined structure and wavelength must be considered for developing LEDs with shorter or longer wave-lengths.
[Comment #5]
In general,the conclusion is not well organized. The results should be further elaborated to show how they could be used for the real applications.
[Response]
Thank you for the suggestion. Please note that we revised the entire conclusion.
Revised manuscript (259~281)
âž The optical efficiency of the 850-nm IR-LED was sharply decreased by a sudden decline in the surface and sideway emission areas of these devices caused by a drop in chip size. In this study, we demonstrated that optical efficiency could be sharply increased by increasing the surface and sideway emission areas through the use of the reflective transparent structure. This is because the surface and sideway emission areas were effectively respectively increased by the transparent and reflective structures in the LED bearing the reflective transparent structure. The transparent epitaxial substrate was secured by applying thick p- and n-AlGaAs epitaxial layers through the LPE method. A reflector bonding structure was sequentially fabricated on the transparent epitaxial structure by applying a reflector and eutectic structure. The presence of the reflective transparent structure in the 850-nm IR-LED was confirmed through SEM. From the schematic of photon paths, the highest optical extraction efficiency was observed from the IR-LED with a reflective transparent structure. This high efficiency was attributed to the enlargement of the effective upward and sideway emission areas caused by the reflective transparent structure. This theoretical result was supported by results of the output power–current–voltage (L-I-V) characteristic and relative photometric measurement. Based on the output power–current (L-I) curves, the IR-LED with reflective transparent structure was found to exhibit a 76% and 26% higher output power than the AS IR-LED and TS IR-LED, respectively. Furthermore, through photometric and radial theta measurements, the emission area of an IR-LED with a reflective transparent structure was much wider and straighter than those of the others. These results suggest that a reflective transparent structure combining the transparent epitaxial structure and reflector bonding structure may considerably increase the optical efficiency of 850-nm IR-LEDs.
[Comment #6]
Could you rewrite the abstract to be more focused on the main goal and the key obtained findings?
[Response]
Thank you for the comment. Please note that we have revised the abstract.
Revised manuscript (14~30)
âž This study investigated a reflective transparent structure to improve the optical efficiency of 850-nm infrared light-emitting diodes (IR-LEDs) by effectively enhancing the number of extracted photons emitted from the active region. The reflective transparent structure was fabricated by combining transparent epitaxial and reflective bonding structures. The transparent epitaxial structure was grown by the liquid-phase epitaxy method, which efficiently extracted photons emitted from the active area in IR-LEDs both to the vertical and horizontal directions. Furthermore, a reflective bonding structure was fabricated using an omnidirectional reflector and a eutectic metal, which efficiently reflected the photons emitted downwards from the active area in an upward direction. To evaluate reflective transparent IR-LED efficiency, a conventional absorbing substrate infrared light-emitting diode (AS IR-LED) and a transparent substrate infrared light-emitting diode (TS IR-LED) were fabricated and their characteristics were analyzed. Based on the power–current (L-I) evaluation results, the output power (212 mW) of the 850-nm IR-LED with the reflective transparent structure increased by 76% and 26% relative to those of the AS IR-LED (121 mW) and TS IR-LED (169 mW), respectively. Furthermore, the reflective transparent structure possesses both transparent and reflective properties, as confirmed by photometric and radial theta measurements. Therefore, light photons emitted from the active area of the 850-nm IR-LED were efficiently extracted upward and sideways because of the reflective transparent structure.

Round 2
Reviewer 2 Report
The authors have carried out a thorough and careful revision and the revised manuscript improved a lot in terms of technical quality and language. Therefore, I would recommend it for publication in the Journal.
Please see the Comments and Suggestions for Authors.